# The Use of Pb Isotope Ratios to Determine Environmental Sources of High Blood Pb Concentrations in Children: A Feasibility Study in Georgia

**DOI:** 10.3390/ijerph192215007

**Published:** 2022-11-15

**Authors:** Adam Laycock, Simon Chenery, Elizabeth Marchant, Helen Crabbe, Ayoub Saei, Ekaterine Ruadze, Michael Watts, Giovanni S. Leonardi, Tim Marczylo

**Affiliations:** 1UK Health Security Agency, Chemical, Radiation and Environmental Hazards Directorate, Harwell Campus, Chilton OX11 0RQ, UK; 2British Geological Survey, Kingsley Durham Centre, Keyworth, Nottingham NG12 5GG, UK; 3UK Health Security Agency, Colindale, London NW9 5EQ, UK; 4The National Center for Disease Control and Public Health, #99 Kakheti Highway, Tbilisi 0198, Georgia; 5Department of Public Health, Environments and Society, London School of Hygiene and Tropical Medicine, London WC1E 7HT, UK

**Keywords:** lead isotope ratio, blood lead concentration, environmental tracing, children, Georgia, lead surveillance, spices, dust, paint, soil

## Abstract

The incidence of lead (Pb) poisoning in children in Georgia has been identified as a major health concern, with a recent national survey identifying that 41% of children aged 2–7 years had blood lead concentrations (BLCs) greater than the blood lead reference value (BLRV) of ≥5 µg dL^−1^. This study collected samples of blood, spices, paint, soil, dust, flour, tea, toys, milk, and water from 36 households in Georgia where a child had previously been identified as having a BLC > BLRV. The Pb concentrations of these samples were determined and compared to Georgian reference values. Samples from 3 households were analysed for their Pb isotope composition. The Pb isotope composition of the environmental and blood samples were compared to identify the most likely source(s) of Pb exposure. This approach identified that some spice and dust samples were the likely sources of Pb in the blood in these cases. Importantly, some soil, paint, and dust sources with high Pb concentrations could be discounted as contributing to blood Pb based on their distinct isotope composition. The data presented demonstrate the significant contribution that Pb surveillance and Pb isotope ratio analyses can make to managing Pb exposure in regions where high BLCs are identified.

## 1. Introduction

Lead (Pb) is a potent neurotoxin and has been a global public health concern for many years [1]. High-level acute exposure can result in damage to the central nervous system, cause a person to enter into a coma, or even result in death, while low level chronic exposure can damage brain development, the central nervous system, the heart, lungs, and kidneys. Chronic Pb poisoning has been associated with a loss of cognition, shortening of attention span, alteration of behaviour, dyslexia, attention deficit disorder, hypertension, renal impairment, kidney damage, immunotoxicity, reproductive toxicity, and increased risk of cardiovascular death [1,2]. Due to the known detrimental health outcomes at any level of Pb, there is currently no level of exposure that is considered to be safe [2].

Children are considered to be at a particularly high risk of the adverse health effects of Pb, especially those under 5 years of age, for several reasons: (i) per unit of body mass, children inhale and ingest more food, water, and dust than adults, so their relative intake of Pb by these routes is therefore greater; (ii) once in the body, children are up to five times more effective at absorbing Pb than adults [2,3]; (iii) hand-to-mouth characteristics are much more prevalent in young children and consequently their ingestion of Pb from sources such as soil, dust, and flaking paint is likely to be considerably greater [3]; (iv) the brains of young children are rapidly developing, but the presence of Pb can interfere with this development and result in long-term irreversible neurological damage [4,5]. Although the negative impacts of Pb poisoning are known, it is estimated to affect hundreds of millions of children globally [1].

Lead exposure is typically assessed by measuring blood lead concentrations (BLCs). As Pb has a half-life of approximately 1–2 months in blood, repeated measurements can be used to monitor changes in exposure with time [6]. Despite there being no safe level of exposure, many public health agencies have blood lead reference values (BLRV) for children. The current BLRV for England and Georgia is 5 µg dL^−1^, while in the U.S., this level has recently been reduced to 3.5 µg dL^−1^ [6,7]. The BLRV is not intended to define an acceptable BLC but rather to identify children of particular concern where public health intervention measures may be implemented and where follow-up monitoring may be required. Indeed, studies have found neurological effects at these levels; for example, at a population level, children with a BLC of 5 µg dL^−1^ have been shown to score several IQ points lower than unaffected peers [8,9].

Concerns about the severity of Pb exposure in Georgia were raised in late 2015 after a collaborative study of 254 children aged 2–5 years at the Lasvili Children’s Hospital in the Georgian capital Tbilisi, performed by Georgia’s National Center for Disease Control and Public Health (NCDC) and U.S. Centers for Disease Control and Prevention [10]. The study found that a third of participants had BLCs above 5 µg dL^−1^. At the same time, the NCDC also conducted a similar study with a cohort of 46 children aged 4–6 years from 10 villages in the Bolnisi and Dmanisi districts, with 14 of the participants (30.4%) having BLCs above 5 µg dL^−1^. The BLC of these participants was retested in 2017–18, with 37% still having BLCs above 5 µg dL^−1^. This led to the UNICEF-funded national Multiple Indicator Cluster Survey (MICS) of Georgia, conducted in 2018 and including a BLC sampling component, where a nationally representative sample of 1578 children aged 2–7 years was tested. It found that, nationally, 41% of participants had BLCs above the BLRV of 5 µg dL^−1^. However, it also demonstrated that there was a significant regional variation, with regions in the west of the country having a higher number of participants with BLCs above the BLRV than those in the east. For example, the region with the lowest proportion of participants having BLCs above the BLRV was Kvemo Kartli (18%), while the highest was Adjara (85%) (Figure 1) [11].

In response, the Georgian authorities instigated ‘The State Program for Disease Early Diagnosis and Screening’ (Ministerial decree No. 240, 23 May 2019, Governmental decree #693, 31 December 2018) in an attempt to mediate the elevated BLCs [12]. This programme included raising public awareness of Pb toxicity and advice on reducing the Pb exposure together with BLC monitoring for MICS participants and their family members under the age of 18. Additionally, children were assessed for physical and mental development and nutritional status. Where BLCs were found to exceed 5 µg dL^−1^, nutritional and behavioural advice was given to the child and their parent/guardian, as well as nutritional supplements of iron, calcium, and multivitamins being prescribed, during a consultation with a doctor. To investigate the impact of these mitigation measures and in an attempt to identify additional potential sources of Pb exposure in these children, a follow-up study was designed by the NCDC, the UK Health Security Agency (UKHSA) (formally Public Health England), and the British Geological Survey (BGS). This study recruited 36 households from across Georgia that had previously participated in the MICS survey and where BLC were previously determined as >5 µg dL^−1^. In addition to the children’s blood, a questionnaire relating to potential Pb exposure and children’s behaviour was completed by the parents/guardians. Based upon the survey responses, a range of samples from the child’s environment were taken where available. The environmental samples included soil, household dust, paint, drinking water, milk, spices, flour, tea, and toys. All samples collected were first analysed to determine the total Pb amount. Samples from three households were also analysed for their Pb isotope composition. Used together, this information can aid in the identification or dismissal of environmental sources of Pb exposure for an individual [13].

The isotope composition of blood Pb is inherited from its source(s); source apportionment can therefore be achieved by comparing Pb isotope ratios in blood with likely sources of exposure. The concept of Pb isotope ratios for environmental tracing has been well established [14,15,16]; however, there are limited applications at a population level. One example of such work was presented by Glorennec et al. [17] and Oulhote et al. [18], where blood samples from 125 French children, along with samples of dust, paint, water, and soil from the child’s household, were analysed for their total Pb concentrations and Pb isotope ratio. In those studies, the observed variability in the isotope ratios and their analytical uncertainty was used to identify that ^207^Pb/^206^Pb and ^206^Pb/^204^Pb were the most appropriate ratios for discrimination in that case. Glorennec et al. [17] and Oulhote et al. [18] reported that 57% of the children had a BLC > 2.5 µg dL^−1^ and of these, a suspected source of Pb exposure was identified in 32% of cases by Pb isotope analysis. In a recent study, similar to the one presented here, by Becker et al. [19], Pb isotope ratios were used to determine that the environmental sources of Pb in five children from Kansas City with BLCs > 5 µg dL^−1^ included household dust, soil, turmeric, and paint. Other studies by Gulson et al. [20] and Takagi et al. [21,22] that used Pb isotope ratios to identify the most likely dominant sources of Pb in the blood of children in Australia and Japan, respectively, also found a strong link with household dust. In a study by Forsyth et al. [23], the blood of 45 pregnant women from a rural area of Bangladesh was analysed and found to be most isotopically similar to that of turmeric, which had elevated levels of Pb and Cr, and indicated that the dominant source of Pb found in blood was from the adulterant spices with lead chromate. Fillion et al. [24] present data from a study of 156 participants from an Inuit population living in Nunavut, Canada, where household members had previously been identified as having a BLC > 10 µg dL^−1^. This study concluded that household dust and ammunition were the likely major sources of Pb exposure. However, as indicated by Gulson et al. [25], some of the ratios presented by Fillion et al. [24] are impossible values and therefore the reliability of the presented data is questionable. Such work highlights the need for robust and reliable methods for data analysis and treatment with the appropriate quality controls and validation.

We aimed to assess the feasibility of applying a Pb isotope ratio tracing approach to the linking of environmental samples as potential sources of Pb exposure in selected children in Georgia, with blood Pb seen as the exposure biomarker and numerous candidate environmental media seen as environmental exposure indicators. We focused on a few of the most exposed children based on total Pb concentration in their blood, but intended to provide an approach for the evaluation of competing environmental sources of exposure in the context of a nationally representative sample. In the following sections, we outline the procedures used for sample preparation and analysis, demonstrate that results are comparable between laboratories, and present preliminary data based on the analyses of samples from three households.

## 2. Materials and Methods

### 2.1. Reagents

Ultrapure water (UPW) from a Milli-Q Reference Water Purification System with a quality of 18.2 MΩ-cm was used throughout. Trace metal grade nitric acid (16 M) and 30% hydrogen peroxide solution for trace analysis were purchased from Fisher Scientific and Sigma-Aldrich, respectively. For total Pb analysis, the following reference materials were included to cover a range of sample types and Pb concentrations: ‘lead in powdered paint’ (Sigma-Aldrich, product ID SQC074), Seronorm Trace Elements in Whole Blood L-3 (SERO), GBW08505 tea leaves, SRM2711a Montana II Soil (NIST) BGS Guidance Material 102 Ironstone Soil (BGS), SRM2584 Trace Elements in Indoor Dust (NIST), and BCR-2 Basalt Columbia River (USGS). Of these reference materials, BCR-2 is the only one with Pb isotope ratio values included on the certificate of analysis. The natural Pb isotope standard SRM981 (NIST) was used as the Pb isotope reference material.

### 2.2. Sample Collection, Transport, and Storage

To focus on participants with the highest levels of Pb exposure, only children identified in the MICS as having BLCs > 10 µg dL^−1^ were considered as potential candidates for this study. This covered 11 of the 13 regions of Georgia (2 regions occupied by Russia were not included), with each region being subdivided into districts. The criteria used to determine if potential candidates should be shortlisted from a given district is outlined in Figure 2. Briefly, where the geometric mean of the BLC for a region was >5 µg dL^−1^, only districts where the arithmetic mean of the BLC was >10 µg dL^−1^ were included. This was the case for 5 of the regions, all in the west of the country. In the 6 regions where the geometric mean of the BLC was <5 µg dL^−1^, only districts where the arithmetic mean of the BLC was >5 µg dL^−1^ were included. The geometric mean was used for regional selection due to the BLC data from the MICS being skewed, with higher incidences of elevated BLCs and the highest BLCs being seen in the west of the country [11]. Any districts that were not selected through the above criteria were still included where soil Pb concentrations, identified through routine state monitoring, exceeded the Georgian reference value of 32 mg kg^−1^ [26]. Prior to the final selection of a household, ethical approval was obtained from the NCDC institutional review board and the parents/guardians were contacted, informed of the aims of the study, invited to be included in the cohort, and provided signed written consent. Thirty-six households were recruited from the shortlisted participants, with a heavier weighting towards the regions and districts with the highest mean BLCs.

For sample collection, a phlebotomist and a trained NCDC scientist visited each household to collect samples and complete a questionnaire with the parents/guardians. In addition to blood samples from the child being taken during the visit, environmental samples of soil, spices, milk, water, household dust, flour (maize and wheat), paint, and toys were taken where available, being potential sources of Pb exposure for the child, and the parent/guardian gave informed consent. The types of environmental samples that were included in this study were selected based on them being reasonable potential sources of Pb, either from the published literature and/or informed by data and information from the Georgian National Food Agency and the Georgian Environment Agency [21,27,28,29].

A single blood sample was taken from the child in all 36 households. The sample was taken from the same child that gave a blood sample for the MICS in all but one instance, where a sibling provided the blood sample. Blood samples of 1–3 mL were collected by a phlebotomist in a vacutainer tube and frozen at −20 °C for storage.

A total of 60 soil and 2 construction sand samples were collected from gardens and yards that the children had access to, and where any vegetables or fruits were grown for household consumption. Approximately 200 g of sample, comprising 10–20 subsamples taken from different locations and a depth of 1–15 cm, was collected. These samples were stored at room temperature.

Aliquots of 2–3 g of the most frequently used spices in the household were sampled, up to a maximum of nine spice types per household. Spices were collected in individual zip lock bags, each labelled with the household reference and spice type. Spices from each household were grouped into plastic zip lock wallets and stored at room temperature.

Approximately 10–20 g of wheat and maize flour used in the household was taken and transferred to labelled zip lock bags and stored in the zip lock wallets with the spices. All households gave at least 1 flour sample, with 61 samples collected in total, including 2 samples of dried corn.

Water and milk samples of between 10 and 50 mL were collected in 50 mL polypropylene tubes and stored at −20 °C. In total, 15 households provided a milk sample sourced from their own domestic supply. All 36 households provided a sample of their main residential drinking water supply, with 17 of these taken from a mains residential water supply and 19 from a private or residential supply from a well or spring. A third (12) of the households also provided a sample from a second source where the child spent a substantial amount of time (e.g., school or family member’s house).

The sampling of household dust followed the procedure outlined by Middleton et al. [30]. Briefly, samples were taken from appropriate raised, flat surfaces where dust is likely to collect, e.g., windowsills, shelves, and tops of wardrobes and cupboards. A plastic template was used to mark out an area 10 cm × 10 cm, then, using Ghost Wipes (Environmental Express, Charleston, SC, USA), the area was sampled and any dust was collected before being transferred to a labelled polypropylene tube. A total of 92 dust samples were collected from 30 households.

Paint samples were collected from 18 households, with 22 samples being collected in total. Samples were taken from both indoor and outdoor locations where painted surfaces were observed as flaking or chipping away. Paint flakes, chips, or dust were collected in labelled zip lock bags.

Where appropriate, toy samples were taken, with a focus on those where the child was known to put the toy in their mouth. In total, 16 toy samples were collected; of these, 12 were plastic toys (e.g., building blocks, animal figures, etc.), 2 were painted metal toys, and 2 were malleable toys (Play-Doh and kinetic sand). These toys were collected in labelled zip lock bags.

Tea samples were gathered from 25 households, with a total of 31 samples being collected, ranging from supermarket black tea bags, herbal teas, and loose tea leaves purchased from local bazaars. A single tea bag or 2–3 g of loose tea leaves was transferred to labelled zip lock bags and stored at room temperature.

### 2.3. Sample Preparation

Blood (1 mL), flour (0.5 g), spice (0.5 g), tea (0.5 g), paint (<0.25 g), and milk (1.5 mL) samples were prepared by microwave-assisted acid digestion. Apart from milk, these samples were prepared in UKHSA laboratories using an Anton Paar Multiwave Go microwave digestion system. Samples were weighed directly into digestion vessels followed by the digestion solution. To the blood samples, 1 mL HNO_3_, 1 mL UPW, and 1 mL H_2_O_2_ were added. To flour, spice, and tea samples, 2 mL HNO_3_, 2 mL UPW, and 1 mL H_2_O_2_ were added. To paint samples, 6 mL HNO_3_ was added. For the blood, flour, spice, and tea sample digests, the temperature was ramped to 180 °C over 10 min and held for 20 min, while for paint samples, the temperature was ramped to 175 °C over 15 min and held for 15 min. Due to restrictions on the importation of dairy products, milk samples were prepared in Georgia by Laboratory Multitest, with 0.5 mL HNO_3_ added and allowed to stand at room temperature for 15 min before the addition of 3 mL UPW. Microwave-assisted digestion was then performed using a Berghof Speedwave Two system, with temperatures ramped to 180 °C over 10 min and held for 20 min. A summary of sample types, amounts, reagents, and digestion protocol used are provided in Table 1. Digested samples were transferred to 15 mL metal-free centrifuge tubes (VWR) and any samples containing undigested material were centrifuged and the supernatant removed. Samples were stored at room temperature prior to analysis.

Water samples were prepared in UKHSA laboratories and 3 mL aliquots were acidified to 5% HNO_3_ and analysed directly. To improve the limits of detection, the remaining samples were preconcentrated using a Genevac EZ-2 Evaporator to reduce sample volumes of up to 50 mL to a final volume of 5 mL in 5% HNO_3_.

To assess the availability of Pb from toy samples, a metal migration protocol was performed as outlined in BSI standard BS EN 71-3:2019 [31]. The toy material was placed into a 0.07 mol L^−1^ HCl solution at a mass ratio of 1:50. The mixture was agitated for 1 h and left to stand for an additional hour. The solid samples were then removed and the solutions acidified to 5% HNO_3_.

Soil samples were prepared in BGS laboratories, with 0.25 g powdered soil weighed into PFA vials. A pre-digest was made by adding 8 M HNO_3_ to the soil and slowly drying down overnight to decompose any organic component. A mixed acid attack consisting of 2 mL 16 M HNO_3_, 2.5 mL HF, and 1 mL 9.2 M HClO_4_ was performed by stepped heating using a programmable heat block. The final heating stage was used to dry down the sample. Once cool, the samples were taken up in 2.5 mL 8 M HNO_3_, 2.5 mL H_2_O_2_, and 20 mL UPW, and transferred to LDPE bottles.

Dust wipe samples were also prepared in BGS laboratories, where they were carefully placed in a disposable DigiTUBE (SCP Science) tube using plastic tweezers. A 10 mL aliquot of 8 M HNO_3_ was added to the sample and left for a minimum of 10 min to allow any initial reaction to die down. A further 4 mL 16 M HNO_3_, 2.5 mL HCl, and 1 mL HF were added before covering with a plastic watch glass and heating overnight at 70 °C on a temperature-controlled hot block. The watch glass was removed and the samples further heated to 110 °C before cooling and the further addition of 4 mL 16 M HNO_3_ and 1 mL H_2_O_2_. A final drying at 90 °C was made until samples achieved a gel-like consistency. The samples were reconstituted with 1 mL 8M HNO_3_ followed by the addition of 9 mL UPW.

### 2.4. Total Pb Concentration Determination

All samples were analysed for total Pb in UKHSA laboratories, where the samples were diluted to 5% HNO_3_ with UPW and any additional dilution was performed with a stock solution of 5% HNO_3_ as required. Analysis was performed using a Thermo iCAP-Q ICP-MS instrument operating in helium collision cell mode and equipped with an ASX-520 autosampler. Calibrations were prepared using standard solutions with Pb concentrations of 0, 0.05, 0.1, 0.5, 1, 5, 10, 50, and 100 µg L^−1^, with a mixing tee used for online internal standard addition of a solution containing 1 µg L^−1^ of Rh. Quality control solutions with Pb of known concentrations were analysed periodically between samples. Quality control solutions of the reference materials SRM2711a, BCR-2, BGS 102, SQC074, and Seronorm Trace Elements in Whole Blood L-3 were analysed between samples, with a summary of this data provided in Appendix A.

All soil and dust samples, and 10 of the tea samples, were also analysed in BGS laboratories, where the sample digests were diluted by a factor of 40 using a mixture of 1% *v*/*v* 16 M nitric acid and 0.5% *v*/*v* 12 M hydrochloric acid. Analysis was performed using an Agilent 8900 ICP-MS instrument operating in helium collision cell mode and equipped with an SPS4 auto-sampler and an ISIS flow injection system. The instrument was calibrated for Pb using 1, 5, 10, 25, 50, and 100 µg L^−1^ dilutions of a custom multi-element mixture from QMX (UK). A quality control solution of SRM2711a, SRM2584, BCR-2, BGS 102, and GBW08505 with a Pb concentration of 25 µg L^−1^ was analysed between samples, with a summary of this data provided in Appendix A.

### 2.5. Pb Isotope Ratio Analysis

Of the 36 households included in this study, the blood and environmental samples from 3 households, hereby referred to as A, B, and C, were selected for Pb isotope ratio analysis. These 3 households were selected based on the types of environmental samples provided and their total Pb concentrations, such that all sample types were included in the isotope analysis and these covered a range of the observed total Pb concentrations.

The sample solutions of blood, paint, spices, tea, flour wheat, and toys were analysed for their Pb isotope composition in UKHSA laboratories. The sample solutions of soil and dust were analysed in BGS laboratories. In addition, four each of the soil and dust samples were analysed in UKHSA laboratories to demonstrate comparability.

The sample analysis and raw data processing performed in UKHSA laboratories were based on the optimised approach developed by Usman et al. [32]. A Thermo Scientific iCAP Q ICP-MS was used for all analyses, measurements were performed in helium collision mode, and the following masses were monitored: ^200^Hg, ^203^Tl, ^204^Pb/Hg, ^205^Tl, ^206^Pb, ^207^Pb, and ^208^Pb. Dwell times of 10 ms were used for each mass, 1000 sweeps per scan were performed, and each analysis consisted of six scans. This resulted in an acquisition time of approximately 7 min, including time for sample uptake and a post-analysis rinse cycle. To ensure that precise and accurate isotope ratios can be determined, it is essential that all masses are monitored in pulse detection mode. When signal intensities reach a certain threshold, the detection mode will switch from pulse to analogue mode; this transition is at approximately 2 × 10^6^ cps. The optimal concentration is therefore low enough for the major ^208^Pb isotope to be monitored without switching to the analogue detection mode while providing a reasonable signal intensity to monitor the minor ^204^Pb isotope.

The deadtime correction factor was established at the start of each measurement session [33]. This was achieved by preparing SRM981 standard solutions at concentrations of 0, 1, 3, 5, 7, and 9 µg L^−1^. With the instrumental deadtime set to zero, these solutions were analysed as above. If the detector switched to analogue mode while monitoring ^208^Pb, the data were omitted. A blank correction was applied to the Pb isotope intensities by subtracting the baseline values from the measured signal intensities (I_blank corr_). The blank corrected intensities were corrected further by changing the dead time (DT), in seconds, using Equation (1).
(1)$$I_{\mathrm{DT}\;\mathrm{corr}}=\frac{I_{\mathrm{blank}\;\mathrm{corr}}}{1-\left(I_{\mathrm{blank}\;\mathrm{corr}}\times \mathrm{DT}\right)}$$

The ^208^Pb/^207^Pb and ^208^Pb/^206^Pb ratios were determined using the blank and dead time corrected intensities (I_DT corr_) plotted against the signal intensity for ^208^Pb. The optimal deadtime value for each isotope ratio is when the slope squared is minimised. This value can be readily found using the Microsoft Excel Solver function. The average optimised dead time for the ^208^Pb/^207^Pb and ^208^Pb/^206^Pb ratios was calculated for the measurement session and set in the instrumental software prior to sample analysis; this ranged from 36–42 × 10^−9^ s across all measurement sessions.

It was determined that the optimal sample Pb concentration for isotope ratio analysis was 7 µg L^−1^. Based on the total Pb concentrations, aliquots of the stock sample digests were prepared at this concentration in 5% HNO_3_ and dosed with Tl to 5 µg L^−1^ with a 1000 µg L^−1^ Tl standard stock solution. Where Pb concentrations in the stock sample solutions were insufficient to achieve a Pb concentration of 7 µg L^−1^, they were analysed without further dilution. Along with the samples, a standard solution of 7 µg L^−1^ SRM981 Pb spiked with 5 µg L^−1^ Tl was prepared in 5% HNO_3_.

The raw signal intensities for each isotope for each run were exported into an Excel file for off-line data processing. They were blank corrected and the following ratios were determined: ^205^Tl/^203^Tl, ^208^Pb/^207^Pb, ^208^Pb/^206^Pb, ^204^PbHg/^208^Pb, ^200^Hg/^208^Pb, ^207^Pb/^206^Pb, ^204^PbHg/^207^Pb, ^200^Hg/^207^Pb, ^204^PbHg/^206^Pb, and ^200^Hg/^206^Pb. Thallium was used for mass bias correction as it is very close in mass to Pb isotopes without having any isobaric interference. Mass bias per unit mass was determined by comparing the measured ^205^Tl/^203^Tl ratio (^205^Tl/^203^Tl_meas_) to the known reference value (^205^Tl/^203^Tl_ref_) of 2.3875. The mass discrimination factor (K) was estimated using the linear law, as shown in Equation (2).
(2)K=T 205l/T 203lrefT 205l/T 203lmeas−1T 205lmass−T 203lmass
where ^205^Tl_mass_ and ^203^Tl_mass_ correspond to the atomic masses of ^205^Tl and ^203^Tl, respectively. The calculated mass bias correction factor for each run was used to calculate mass bias corrected isotope ratios (IR_corr_) for each of the measured isotope ratios (IR_meas_) using Equation (3).
(3)$${\mathrm{IR}}_{\mathrm{corr}}={\mathrm{IR}}_{\mathrm{meas}}\left(1+\left(K\Delta m\right)\right)$$
where Δm is the difference in the atomic mass between the two isotopes in the ratio. The mass bias corrected ^200^Hg/^20x^Pb ratios (where ‘^x^’ represents 6, 7, or 8) were then used, with the known ^204^Hg/^200^Hg reference value (^204^Hg/^200^Hg_ref_) of 0.2974, to correct for the isobaric interference of ^204^Hg on the ^204^PbHg/^20x^Pb mass bias corrected ratios (^204^PbHg/^20x^Pb_corr_) to derive the mass bias corrected ^20x^Pb/^204^Pb ratios (^20x^Pb/^204^Pb_corr_).
(4)P 20xbP 204bcorr=PbHg 204P 20xbcorr−H 200gP 20xbcorr×H 204gH 200gref−1

The SRM981 Pb standard was analysed at least once every 10 samples. For each standard analysis, the deviation in the mass bias corrected ratio from the true reference ratio was determined. Linear interpolation of the deviation in the SRM981 Pb standard was used to apply this correction to samples analysed over the measurement session to derive the final, true isotope ratios. The analytical uncertainty was derived by calculating twice the standard error of the mean of the six scans for each of the isotope ratios.

The Pb isotope analysis performed in BGS laboratories used a comparable approach as outlined above with a few small differences. Analysis was performed using an Agilent 8900 ICP-MS instrument equipped with an SPS4 auto-sampler and an ISIS flow injection system. Samples were diluted by an individual factor calculated from Pb concentrations and instrument sensitivity to give a ^208^Pb^+^ count rate of c. 1 × 10^6^ cps to optimise counting statistics, within the pulse counting range of the detector c. 1.6 × 10^6^ cps. The isotopes ^200^Hg^+^, ^204^Pb^+^, ^206^Pb^+^, ^207^Pb^+^, and ^208^Pb^+^ were acquired with dwell times to give 1000 sweeps per integration and 10 integrations over 5 min. Thallium was not used to correct mass bias as the BGS has previously found that this showed no improvement over only standard-10 × sample-standard bracketing. Quality control was provided by repeatedly analysing an in-house UK ore Pb (Glendenning) throughout the runs for which the BGS has more than five years of isotope ratio data, including from high-precision multi-collector ICP-MS analysis. A summary of the quality control analysis is provided in Appendix A.

## 3. Results and Discussion

### 3.1. Interlaboratory Comparison

The total Pb concentration of the soil (*n* = 60), sand (*n* = 2), dust (*n* = 92), and 10 of the tea samples, a total of 164 samples, was determined by both the UKHSA and BGS laboratories. Overall, 90% of the concentrations determined by the two laboratories agreed within 10% and 99% of samples agreed within 20%. The total Pb concentration determined by the laboratories for two of the dust samples did not agree within 20% of each other. One of these was the dust sample with the highest total concentration of Pb, and so the higher dilution factors used to prepare the sample for analysis may have introduced some additional error. Appendix A shows that the comparability of Pb concentration results between laboratories is very good. The results (not reported here) from an initial model indicated a serious violation of the model assumptions, in particular, the variance and mean relationship assumption. Therefore, we needed a modelling approach that would take into account the dependence of variance to the mean. However, the modelling results using a log transformation of the lead concentration were satisfactory and the model diagnostic did not show any serious model assumptions failure. Therefore, we reported the model results from modelling the log-transformed lead concentration. The results indicated a significant lead concentration difference between the two labs. The lead concentration reported by the UKHSA was higher than that of the BGS. The lead concentration geometric mean from the UKHSA was exp(0.02) = 1.02 for soil, exp(0.04) = 1.04 (dust), and exp(0.02) = 1.02 (tea) times that of the BGS, respectively. A similar conclusion was obtained by using the Wilcoxon Signed-Rank test.

For comparison of the isotope ratio data, four soil and four dust samples were analysed by both laboratories (Appendix A). It can be seen that the analytical uncertainty achieved in the laboratories is comparable and that the results from each lab are within this uncertainty, with the exception of one of the soil samples where the ^20x^Pb/^204^Pb ratios determined by the UKHSA were higher than those from the BGS laboratory. A Shapiro–Wilk normality test showed that the difference in ratios between laboratories followed a normal distribution, while a paired two-tailed t-test showed that there was no difference between the isotope ratio data determined by the UKHSA and BGS laboratories at a significance level of 0.05.

### 3.2. Pb Concentrations

The BLC of the 36 participants ranged between 2.6 and 39.9 µg dL^−1^, with all but two of the samples being greater than or equal to the Georgian BLRV of 5.0 µg dL^−1^ (Table 2 and Figure 3). The two participants with BLCs below the BLRV were both from the Kvemo Kartli region in the east of the country, while the highest BLCs were seen in the southwest of the country in the Adjara, Imereti, and Guria regions. This observation is in line with the regional BLC trends identified in the MICS and subsequent state programme surveys [11,34].

The spice samples were grouped into 11 categories: spice mixes (20), coriander (16), blue fenugreek (14), paprika (13), saffron/yellow flower (13), black pepper (13), seasoned salt (11), rosemary (10), adjika (8), cumin (8), whole chilli (6), and others (6), with the numbers in parentheses showing the total number of samples collected for each category. The ‘spice mixes’ category had the most samples and included various mixes for seasoning meat or fish; kharcho spice mixes, a seasoning used in a range of Georgian dishes; sacivi seasoning, used in a traditional Georgian dish of poultry in a walnut sauce; and mixes of coriander with other spices and seasoning. The ‘others’ category included the least common spice types collected, such as ginger, pennyroyal, garlic, parsley, and basil. The spice samples had a greater variability in total Pb concentrations than any other sample type and were in the range of 0.01–6165 mg kg^−1^. The highest concentrations were observed in particular types of spices, including spice mixes, blue fenugreek, and saffron (Figure 4, Table 3). These spices have previously been identified as having high concentrations of Pb in studies investigating spices sold in Georgia and those imported into the U.S. from Georgia and other countries [27,28,35]. The reference value for Pb in spices in Georgia is 5 mg kg^−1^; of the 136 spice samples analysed here, 59 (43%) exceeded this value. In 6 of the 12 spice categories, it was found that more than half the samples were above the reference value, with blue fenugreek and adjika having the highest proportion of samples, 79% and 75%, respectively, above the reference value. Even in spice categories where the mean Pb concentration was below the reference value, such as coriander (1.4 mg kg^−1^), rosemary (1.9 mg kg^−1^), and black pepper (0.4 mg kg^−1^), up to 30% of the samples had a Pb concentration above the reference value up to 22.2 mg kg^−1^. Although this is more than four times above the reference value, it is approximately ten times lower than the highest Pb concentrations seen in the other spice sample categories (Table 3). The only category with Pb concentrations consistently below the reference values was fresh chillies, which were generally sampled from households’ vegetable patches. The high Pb concentrations seen in some of these samples may be a consequence of contamination from inappropriate or poorly maintained processing equipment or accumulated from high levels of Pb in the soil where the spice was originally grown. The adulteration of spices is also a documented practice, whereby pigments such as lead chromate are added to the spice in order to give it a more vivid colour and therefore make a product visually appealing to consumers and/or to increase the density of spices that are sold by weight [28,36]. Previous studies have linked spices adulterated with lead chromate to blood Pb by using total Pb concentrations and lead isotope ratios [23].

Leaded paint is a well-documented source of Pb exposure and as such, has been banned in many countries. The present study analysed 22 paint samples that were shown to have Pb concentrations in the range of 1–4800 mg kg^−1^ (Table 2, Figure 3). Of the 22 samples, 12 (55%) were above the Georgian reference value (90 mg kg^−1^) for Pb in paint, a larger proportion than any other sample type included in this study, with 6 (27%) of the samples being above 600 mg kg^−1^. This compares well with a report by the International Pollutants Elimination Network which identified a 2016 Russian study of 37 paints sold in Georgia that found that 4 (11%) paints had Pb concentrations above 11,000 mg kg^−1^ and 12 (32%) had concentrations above 600 mg kg^−1^, with the maximum reported being 68,000 mg kg^−1^ [29].

The soil samples showed a relatively limited range in Pb concentrations of 9–158 mg kg^−1^. Of the 60 soil samples, 16 were found to be above the Georgian reference value for Pb in the soil (32 mg kg^−1^). The 2 construction sand samples had concentrations of 10.9 and 12.8 mg kg^−1^. The concentration range seen in the soils here is similar to that reported in the Geochemical Mapping of Agricultural soils of Europe (GEMAS) and Forum of European Geological Surveys (FOREGS) consortia, which demonstrate that agricultural soils within Europe typically lie in the range of 1.6–100 mg kg^−1^, with a maximum value of 1309 mg kg^−1^ [37,38]. However, in urban regions and locations close to Pb mineralisation, much higher concentrations are not unexpected. For example, a British Geological Survey report for the UK Department for Environment, Food and Rural Affairs on normal levels of contaminants in English topsoil reported Pb concentrations ranging from 2–10,200 mg kg^−1^, with the normal background concentrations (NBC) (defined as the upper 95% confidence limit of the 95th percentile of the domain of interest) for urban and mineralisation (soils that have been impacted by Pb mineralisation with a history of mining and associated activities) domains being 820 and 2400 mg kg^−1^, respectively [39]. All other areas not covered by these domains were reported to have an NBC of 180 mg kg^−1^ [40]. There is no single safe limit threshold for Pb in soil; however, the U.S. Environment Protection Agency recommend that areas occupied by children should not have concentrations exceeding 400 mg kg^−1^ [41].

The Pb content of the 92 household dust samples were in the range of 4 to 2163 µg m^−2^; the Georgian reference value for dust is 431 µg m^−2^, but 9 of the 92 samples exceeded this value. As the total amount of dust sampled is unknown, it is not possible to determine the mass/mass concentration of Pb in the dust; instead, the mass per unit area sampled is reported. These values will therefore be largely dependent on the accumulation rate of dust in the sampling region and the time over which dust has accumulated, and this may explain much of the variability observed in the Pb concentrations. The Pb concentrations in dust compare well with those found in previous studies; for example, Fillion et al. [24] reported Pb concentrations in household dust of 1.9–3491 µg m^−2^ from an Inuit population in Nunavut, Canada, while from 24 households in Australia, Gulson et al. [20] measured household dust in the range of 1.4–279 µg m^−2^. The latter study collected dust samples from floors, as opposed to surfaces that are more likely to accumulate dust, as was performed in this study and which may in part explain the higher concentrations seen in this study.

The Pb concentrations of the dried tea leaf samples were in the range of 0.07–2.11 mg kg^−1^ (Figure 3, Table 2), below the Georgian reference value for Pb in tea of 10 mg kg^−1^. Of the 31 tea samples analysed, 3 were >1 mg kg^−1^. These results are comparable to a 2015 report from the UK Food Standards Agency for which the concentration of Pb in 62 dried tea leaf samples ranged from 0.13–2.56 mg kg^−1^, with liquid tea brewed under different conditions presenting the majority of samples below the limit of quantification of 0.7 µg L^−1^ [42]. In 2014, a safety limit of 1 mg kg^−1^ for ‘the dried leaves and stalks, fermented or otherwise of *Camellia sinensis*’, which encompasses black tea, green tea, and white tea, was tentatively proposed by the European Commission. However, due to a lack of evidence, the Regulation EC No 1881/2006 ‘setting maximum levels for certain contaminants in foodstuffs’ has not been modified to adopt this limit [43].

Only a small proportion of samples of flour, milk, toys, and water were found to contain enough Pb to perform isotope ratio analysis. Of the 60 flour samples, 55 had total Pb concentrations of <0.03 mg kg^−1^, while the remaining 5 were in the range of 0.07–1.49 mg kg^−1^. All of the water and milk samples were found to be less than 0.03 and 7 µg L^−1^, respectively.

### 3.3. Pb Isotope Ratios

The Pb isotope ratio data are presented in Figure 5 within the context of each household (A–C). The isotope signature of blood Pb is inherited from the environmental source(s) from which the Pb originated. Therefore, if there is a single dominant source of Pb in the blood, the isotope signature will be indistinguishable from that of the source. Here, we define sources as being isotopically indistinguishable if their isotope ratios are within the analytical uncertainty (calculated as twice the standard error of the mean) of the measurement. If there is a significant contribution of Pb in the blood from a number of sources, the Pb isotope signature of the blood shall be some intermediate of these and determined by their relative contributions [21,44].

Household A included environmental samples of dust (3), soil (3), spice (7), paint (1), tea (4), and water (1), and the participant’s BLC was 14.0 µg dL^−1^. For this household, the water, soil, and tea samples’ Pb concentrations were low and below levels that may indicate concern. In this case, the soil samples were the most isotopically distinct from blood Pb and were also distinct from the other Pb sources (Figure 5A). The tea and water samples were more isotopically similar to each other but also distinct from blood Pb. The household dust, spice, and paint samples provided from this household all included individual samples with Pb concentrations greater than the reference values. The paint sample was from a window frame and had a Pb concentration of 438 mg kg^−1^. The isotope composition of the Pb was, coincidently, comparable to the tea samples and distinct from blood Pb. The three household dust samples had Pb concentrations that were amongst the highest seen in the study, at 583, 1607, and 2163 µg m^−2^, and were isotopically similar but distinct from each of the other sample types. This indicated that Pb in the soil and from paint is not a significant contributor to Pb in the dust, and that none of these are significant contributors to blood Pb. The Pb concentrations in the seven spice samples ranged from 0.3–6165 mg kg^−1^, four of which were above the reference value of 5 mg kg^−1^. Three of these samples were isotopically indistinguishable from each other and blood Pb. Blood Pb was isotopically indistinct from three of the spice samples (coriander—20.4 mg kg^−1^, blue fenugreek—618 mg kg^−1^, and kharcho—6165 mg kg^−1^), all of which were above the reference value, suggesting that there was a strong likelihood that the dietary uptake of Pb from spices was the predominant source of Pb in the blood. This was further supported by the fact that of the samples analysed from this household, the spice and blood Pb samples showed the highest values in terms of their ^208^Pb/^206^Pb and ^207^Pb/^206^Pb ratios, and therefore the isotope signature in blood Pb could not be a result of a mixing of multiple sources of the environmental samples analysed.

Household B included environmental samples of dust (3), flour (2), milk (1), soil (2), spice (5), paint (1), toy (1), and water (2), and the participant’s BLC was 8.7 µg dL^−1^. Of these samples, the flour, milk, paint, toy, and water samples were found to have low Pb concentrations. However, there were sufficient concentrations of Pb in the water, toy, and paint samples for isotope analysis, and these were found to be isotopically distinct from each other and blood Pb. One of the soil samples from this household, with a concentration of 58 mg kg^−1^_,_ was above the Georgian reference value of 32 mg kg^−1^, and although it was isotopically similar to blood Pb, it was still distinguishable from it (Figure 5B). The three dust samples from this household had Pb concentrations of 60, 92, and 280 µg m^−2^, all of which are below the Georgian reference value of 430 µg m^−2^. The dust sample with the highest Pb content, taken from the living room floor, was found to be isotopically indistinct from blood Pb, while the other dust samples, taken from high surfaces in the kitchen and the child’s bedroom, were isotopically indistinguishable from each other but distinct from blood Pb. Previous work by Middleton et al. [30] demonstrated that sampling location is likely important for both elemental loading, whereby higher levels may be expected close to entrances relative to interior rooms and a representative reflection of human exposure; for example, floors may be more representative of a child’s exposure than difficult-to-reach locations. This may explain why the dust sample from the living room floor has a higher Pb concentration and is isotopically indistinct from blood Pb. Two of the five spice samples had Pb contents above the Georgian reference value of 5 mg kg^−1^; with concentrations of 305 and 5321 mg kg^−1^, these were blue fenugreek and kharcho, respectively. Of these, the former is isotopically indistinguishable from blood Pb and the latter is isotopically distinct. This indicates that while the kharcho sample has a Pb concentration 1000 times greater than the reference value, it does not seem to be a significant source of blood Pb. A second spice sample, basil, with a much lower Pb concentration of 4.8 mg kg^−1^, was also isotopically indistinct from the blood. In this case, the most likely source of Pb is from the blue fenugreek.

Household C included environmental samples of dust (3), flour (2), milk (1), soil (2), spice (6), tea (1), and water (2), and the participant’s BLC was 18.6 µg dL^−1^. The Pb concentrations in the flour, water, and milk samples were below or close to the limits of detection and were therefore not analysed for their Pb isotope composition. The tea and soil samples were below Georgian reference values and were isotopically distinct from blood Pb (Figure 5C). The three household dust samples had comparable concentrations of Pb at 169, 173, and 204 µg m^−2^. Of the three dust samples, the one taken from the kitchen is isotopically indistinct from blood Pb, and while the other two samples, taken from the living room and the child’s bedroom, are isotopically similar, they are still distinct. However, if dust was the dominant source of Pb, and all three dust sources contributed equally, the resulting isotope composition would be indistinct from blood Pb. Of the five spice samples analysed, three were found to have Pb contents above the Georgian reference value, with concentrations of 10, 42, and 212 mg kg^−1^ corresponding to paprika, blue fenugreek, and saffron, respectively. As with household B, the spice with the highest concentration was isotopically distinct from blood Pb but the blue fenugreek was indistinct, indicating that this is a contributor to blood Pb.

## 4. Conclusions

In this study, 34 of the 36 participants (94%) were found to still have BLCs above the BLRV, indicating that Pb exposure continues to be a public health concern in Georgia. The Pb concentration data for the environmental samples showed that the sample types with the highest proportion of samples above their corresponding reference values were paint (55%), spice (43%), soil (25%), and dust (10%). Of these, paint and spice samples were seen to exceed reference values by the greatest degree, in the worst cases by more than 1200 times. A comparison of the Pb isotope compositions of the samples from the three selected households showed that Pb in the paint and soil was isotopically distinguishable from Pb in the dust and blood. This indicates that soil and paint do not make a significant contribution of Pb to dust or blood. It was also found that the Pb in some of the dust and spice samples were isotopically indistinct from the blood. This provides a strong indication that the Pb from dust and spices made a significant contribution to the Pb in the blood. This finding is in agreement with other studies where an association with household dust and blood Pb has been suggested [19,20,21,22,24]. The Pb isotope data showed that it was not always the environmental samples with the highest total Pb concentrations that were the likely main contributory sources of blood Pb. For example, two of the households provided a kharcho spice sample, both of which had some of the highest concentrations of total Pb seen for that sample type; however, the Pb isotope data indicated that in only one of these instances was this a likely source of blood Pb. This analysis was able to identify and discount sources of Pb to the blood on a case-by-case basis and can be used as a powerful tool to guide intervention advice and reduce an individual’s exposure to Pb. This feasibility study demonstrates that the methods presented produce robust, reliable, and reproducible lead isotope ratio data, and they were able to identify potential sources of Pb exposure in the three households that were investigated here. Expanding this approach to a wider range of participants in Georgia is required to provide more of a nationally representative overview of Pb exposure for Georgian children.

## Figures and Tables

**Figure 1 ijerph-19-15007-f001:**
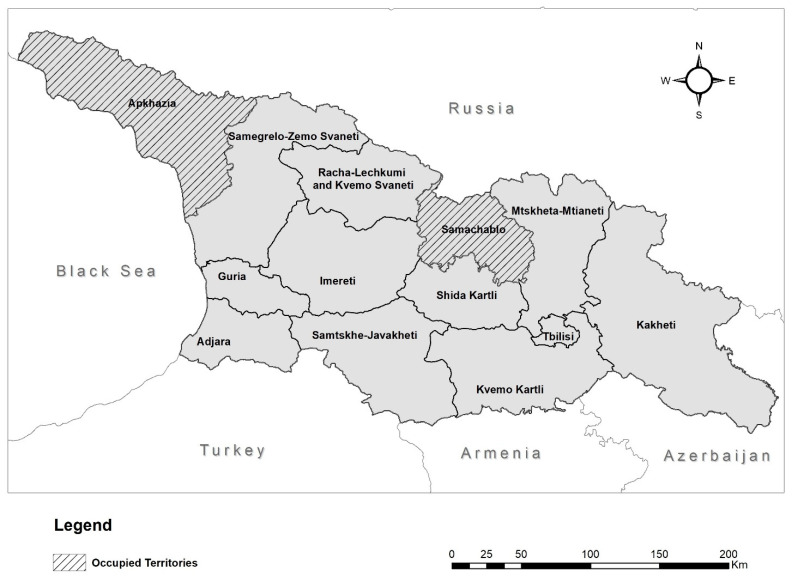
Map of Georgia showing its borders with neighbouring countries and the regions within Georgia.

**Figure 2 ijerph-19-15007-f002:**
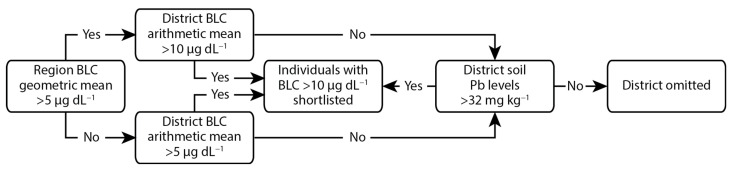
Summary of the criteria used to identify potential candidates based on data from the MICS.

**Figure 3 ijerph-19-15007-f003:**
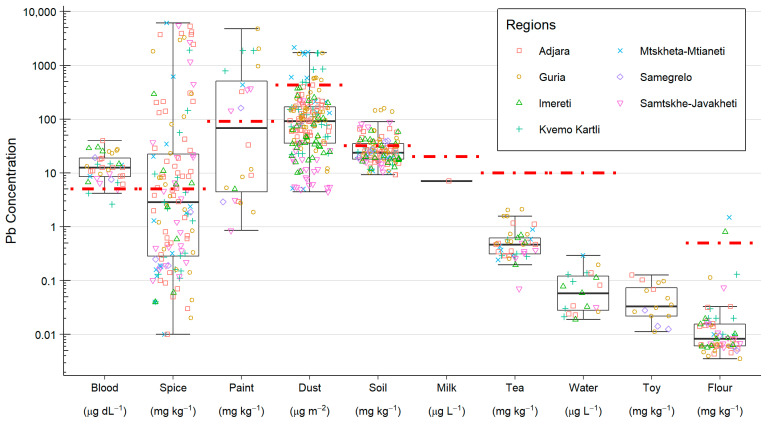
Box and whisker plot of the Pb concentrations for all samples. Where applicable, Georgian reference values are indicated by a horizontal red-dashed line.

**Figure 4 ijerph-19-15007-f004:**
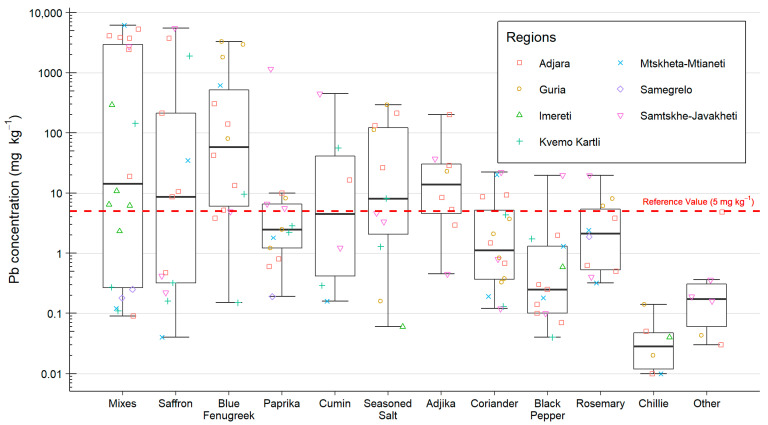
Box and whisker plot of the Pb concentration for each spice category analysed in this study. The plot shows the mean, median, and interquartile range of the data, and any outlier samples. The Georgian reference value for Pb in spice (5 mg kg^−1^) is indicated by the red-dashed line.

**Figure 5 ijerph-19-15007-f005:**
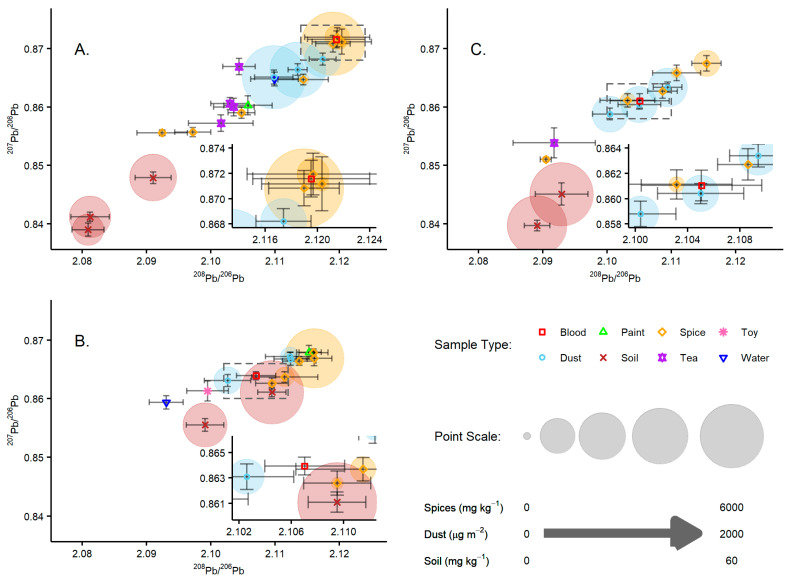
Lead isotope plots of the blood and environmental samples collected from each of the three households. Insets show a zoomed-in area of the plots focused around the corresponding blood data and are indicated by the rectangles outlined by a dashed line in the main plot. Scaled points are also included for the spice, dust, and soil samples to give an indication of the total Pb concentration for the corresponding sample. (**A**)—Isotope ratio data from Household A. (**B**)—Isotope ratio data from Household B. (**C**)—Isotope ratio data from Household C.

**Table 1 ijerph-19-15007-t001:** Summary of microwave-assisted acid digestion protocols used for sample digestion.

Sample	Blood	Flour, Spice and Tea	Paint	Milk
Volume/Mass	1 mL	0.5 g	<0.25 g	1.5 mL
Laboratory–Instrument	UKHSA–Anton Paar Multiwave Go			Multitest–Berghof Speedwave Two
Reagents	1 mL HNO_3_	2 mL HNO_3_	6 mL HNO_3_	0.5 mL HNO_3_
	1 mL UPW	2 mL UPW		3.0 mL UPW
	1 mL H_2_O_2_	1 mL H_2_O_2_		
Digestion protocol	Ramp to 180 °C over 10 min, hold for 20 min		Ramp to 175 °C over 15 min, hold for 15 min	Ramp to 180 °C over 10 min, hold for 20 min

**Table 2 ijerph-19-15007-t002:** Summary of Pb concentration data as determined for each sample type.

Sample	Units	*n*	Min	Max	Percentile		Geometric Mean	Arithmetic Mean	Median	Reference Value ^a^	>Reference Value (*n*)
					25th	75th					
Blood	µg dL^−1^	36	2.6	39.9	8.5	18.7	12.2	14.5	12.6	5	34 [94%]
Paint	mg kg^−1^	22	0.8	4802	4.1	699	73.0	644	153	90	12 [55%]
Spice	mg kg^−1^	136	0.01	6164	0.3	22.4	4.1	387	2.9	5	59 [43%]
Soil	mg kg^−1^	62	10.5	158.8	17.9	35	25.8	31.7	23.4	32	16 [26%]
Dust	µg m^−2^	92	34.9	2163	34.9	168	79.5	198	88.6	431	9 [10%]
Tea	mg kg^−1^	31	0.07	2.1	0.3	0.6	0.4	0.5	0.4	10	0 [0%]
Flour	mg kg^−1^	62	<0.01	1.5	0.01	0.02	0.01	0.05	0.01	0.5	2 [3%]
Water	µg L^−1^	48 ^b^	0.02	0.3	0.03	0.12	0.1	0.1	0.1	10	0 [0%]
Milk	µg L^−1^	16 ^c^	7.1	7.1	7.1	7.1	7.1	7.1	7.1	20	0 [0%]
Toys	mg kg^−1^	16	11.2	126	21.8	74.3	37.8	49.9	33.1	-	-

^a^ Georgian reference values refer to the maximum permissible concentrations as determined by Georgian ministerial decrees, ^b^ 19 samples above the LOQ, ^c^ 1 sample above LOQ.

**Table 3 ijerph-19-15007-t003:** Summary of spice Pb concentrations.

Spice Type	*n*	Lead Concentration mg kg^−1^							>5 mg kg^−1^ (*n* [%])
		Median	Percentile		Min	Max	Geometric Mean	Arithmetic Mean	
			25th	75th					
Spice mixes	20	14.9	0.3	2971	0.09	6165	28.7	1444	13 [65%]
Saffron/yellow flower	13	8.7	0.3	211	0.04	5510	8.9	879	7 [54%]
Blue fenugreek	14	61.3	6.3	540	0.15	3292	55.9	664	11 [79%]
Paprika	13	2.5	1.2	6.6	0.19	1160	3.5	92.5	5 [38%]
Cumin	6	8.8	0.5	46.1	0.16	449	5.3	87.1	3 [50%]
Seasoned salt	11	8.1	2.3	122.1	0.06	295	9.1	72.4	6 [55%]
Adjika	8	15.7	4.7	30.7	0.45	202	11.4	39	6 [75%]
Coriander	16	1.2	0.4	5.4	0.12	22.2	1.4	4.7	4 [25%]
Black pepper	13	0.3	0.1	1.3	0.04	19.8	0.4	2.0	1 [8%]
Rosemary	10	2.1	0.5	5.5	0.32	19.6	1.9	4.4	3 [30%]
Chillies	6	0.0	0.0	0.0	0.01	0.1	0.0	0.0	0 [0%]
Other	6	0.2	0.1	0.3	0.03	4.8	0.2	0.9	0 [0%]
All	135	2.9	22.6	943	0.01	6165	4.2	389	59 [44%]

## Data Availability

The data presented in this study are available on request from the corresponding author.

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
