# Peer review of "The Use of Pb Isotope Ratios to Determine Environmental Sources of High Blood Pb Concentrations in Children: A Feasibility Study in Georgia"

_ijerph, 2022, doi:10.3390/ijerph192215007_

Round 1

Reviewer 1 Report

Interesting and relevant study since Pb exposure continues to be a public health concern in Georgia.

Results may become a powerful tool to guide intervention advice and reduce an individual’s exposure to Pb (knowledge transfer is a great outcome of this manuscript).

Introduction and all other sections are well written. 

The paper seems to be too long so just a few suggestions in this sense: Is really Figure 1 necessary?

The criteria used to identify potential candidates are well presented. Quality control analysis is correct.

The selection of the different food products (tea, milk and species) seems to be ramdon. Authors are invited to better justify the selection of these food groups and why not others. Tea and species are expected to be food groups with very little consumption rates so why testing them?. The Pb dietary exposure from them may contribute very little to the total dietary intake. Have they been tested beceause they are locally produced or because they are an important industry in Georgia?

Water is expected to be analysed and it is not necessary to justify it.

About the Georgian reference values, what is the reference? Does it only include the above mentioned and tested food groups or also other ones?. Will this paper suggest the inclusion of new food groups in that reference values list?

Are authors consistent with the following statement (line 450): "The intentional addition of Pb would also increase the density of spices, which may be a motivation when products are sold by weigh". Isn`t this statement too much without evidences?

Is really Figure 4 and its content and discussion of interest for the international community reading this paper?. Why not just reduce the discussion to a national prespective concluding with recommendations to regulators and Pb monitoring responsibles?

Author Response

See attached word document

Author Response

See attached word document

Reviewer 3 Report

Manuscript Number: ijerph-1999583

Title: Pb isotope ratios to link environmental sources with high blood Pb concentrations in children: a feasibility study in Georgia

The manuscript provided appropriate information about the studied task, but there are several requirements that have to consider by the authors. In this regard, the following comments are requested to be addressed by the authors.

       Abstract

Samples from 3 households are representative enough?

Key words

Authors should rephrase keywords. Do not use words or terms in the title as keywords: the function of keywords is to supplement the information given in the title. Words in the title are automatically included in indexes, and keywords serve as additional pointers.

 Introduction

The background and motivation should refined and this section can add more descriptions of the specific work in the following text. For example, the source of lead pollution and the disposal.

Line 52-53 lack of citation.

 Materials and Methods

Reorganizing the language can make the content more coherent.

Proper reference is required.

Result and Discussion

Interpretation of the result and conclusions need to be focused.

Guidance can be added as appropriate based on the results of the analysis to echo the outlook section of the conclusion.

 Conclusion

Lack of more valuable summative work.

Reference

Some references are too old.

Author Response

See attached Word document
